# In Silico Mixed Ligand/Structure-Based Design of New CDK-1/PARP-1 Dual Inhibitors as Anti-Breast Cancer Agents

**DOI:** 10.3390/ijms241813769

**Published:** 2023-09-06

**Authors:** Alessia Bono, Gabriele La Monica, Federica Alamia, Francesco Mingoia, Carla Gentile, Daniele Peri, Antonino Lauria, Annamaria Martorana

**Affiliations:** 1Dipartimento di Scienze e Tecnologie Biologiche Chimiche e Farmaceutiche “STEBICEF”, University of Palermo, Viale delle Scienze, Ed. 17, 90128 Palermo, Italy; alessia.bono01@unipa.it (A.B.); gabriele.lamonica01@unipa.it (G.L.M.); federica.alamia0095@gmail.com (F.A.); carla.gentile@unipa.it (C.G.); annamaria.martorana@unipa.it (A.M.); 2Istituto per lo Studio dei Materiali Nanostrutturati (ISMN), Consiglio Nazionale delle Ricerche (CNR), 90146 Palermo, Italy; francesco.mingoia@ismn.cnr.it; 3Dipartimento di Ingegneria dell’Innovazione Industriale e Digitale, Università degli Studi di Palermo, Viale 10 delle Scienze Ed. 6, 90128 Palermo, Italy; daniele.peri@unipa.it

**Keywords:** breast cancer, CDK-1, PARP-1, *olaparib*, *dinaciclib*, multitarget mechanism, DRUDIT, NCI database

## Abstract

CDK-1 and PARP-1 play crucial roles in breast cancer progression. Compounds acting as CDK-1 and/or PARP-1 inhibitors can induct cell death in breast cancer with a selective synthetic lethality mechanism. A mixed treatment by means of CDK-1 and PARP-1 inhibitors resulted in radical breast cancer cell growth reduction. Inhibitors with a dual target mechanism of action could arrest cancer progression by simultaneously blocking the DNA repair mechanism and cell cycle, resulting in advantageous monotherapy. To this aim, in the present work, we identified compound **645656** with a significant affinity for both CDK-1 and PARP-1 by a mixed ligand- and structure-based virtual screening protocol. The Biotarget Predictor Tool was used at first in a Multitarget mode to filter the large National Cancer Institute (NCI) database. Then, hierarchical docking studies were performed to further screen the compounds and evaluate the ligands binding mode, whose putative dual-target mechanism of action was investigated through the correlation between the antiproliferative activity data and the target proteins’ (CDK-1 and PARP-1) expression pattern. Finally, a Molecular Dynamics Simulation confirmed the high stability of the most effective selected compound **645656** in complex with both PARP-1 and CDK-1.

## 1. Introduction

Breast Cancer (BC) is one the most common cancers in women worldwide (excluding nonmelanoma skin cancers), causing about 40,000 deaths per year [1,2,3]. BCs can be categorized in three major BC subtypes: hormone receptor positive/ERBB2 negative (HR+/ERBB2−) [4,5], ERBB2 positive (ERBB2+) [6,7], and triple-negative (TN) [8,9,10].

Approximately 5–10% of BC cases follow a Mendelian (autosomal dominant) inheritance pattern, while 15/20% of cases are familial, among which at least 30% are attributed to germline mutations in the *BRCA1* and *BRCA2* genes [11].

From a medical point of view, for nonmetastatic BC, eradicating tumors from the breast and regional lymph nodes and preventing metastatic recurrence through surgical resection represent the main goals of therapy.

Systemic treatments may be preoperative (neoadjuvant), postoperative (adjuvant), or both. The BC subtype guides the standard systemic therapy administered, which consists of endocrine therapy (*tamoxifen* [12,13,14], *letrozole* [15,16,17,18], *anastrozole* [19], *exemestane* [20,21]) for all HR+ tumors, trastuzumab-based ERBB2-directed antibody therapy plus chemotherapy for all ERBB2+ tumors [9] (such *cyclophosphamide* [22,23], *paclitaxel* [22,24,25], *docetaxel* [24], *carboplatin* [26,27,28]), and classical chemotherapy alone for triple-negative breast cancer (*capecitabine* [29,30,31], *eribulin* [32,33], *vinorelbine* [34,35], *gemcitabine* [36,37], *talazoparib* [38]).

On the other hand, for Metastatic Breast Cancer (MBC), therapeutic goals are prolonging life and symptom palliation with available agents [39,40]; however, there is currently no “gold standard” in this setting and chemotherapy for MBC (including TN and negative for estrogen and progesterone receptors) has become increasingly complex.

From this point of view, complementing existing therapies with adjuvant agents able to interact with targets different from estrogen and progesterone receptors could represent a key alternative in the MBC scenario. Indeed, to date, several clinical investigation studies have been focused on drugs acting against targets that are overexpressed in BC cells, among which, particular interest is committed to CDK-1 and PARP-1.

PARP-1 is one of the main participants in DNA repair, playing a key role in terms of Base Excision Repair (BER) and DNA Single-Strand Break (SSB) repair [41]; thereby emerging as an attractive target in anti-cancer drug discovery projects.

In particular, BC cells have been reported to be significantly reliant on DNA repair pathways and are therefore susceptible to DNA-damage response inhibition [42]. Indeed, preclinical data have revealed that BRCA1/2-mutant cancer cells are sensitive to PARP-1 inhibition due to their dependence on PARP-1 activity for DNA (base excision) repair and, subsequently, survival [43,44,45,46,47,48,49,50,51].

CDK-1, which is crucial in centrosome regulation, can form a complex with CyclinB1 and control entry into mitosis [52,53,54], enhance chromosome condensation, and nuclear envelope breakdown [54,55,56]. Previous research found that CyclinB1 and CDK-1 are highly expressed in BC cells and are associated with patients’ overall survival [57,58,59,60,61,62,63,64,65]

In this light, *dinaciclib* (MK-7965, formerly SCH727965) and *olaparib* (Lynparza^®^), CDK-1 and PARP-1 inhibitors, respectively, are two of the most potent small molecules with nM IC_50_ values in vitro (Figure 1).

Olaparib has been the first PARP inhibitor with reported positive results from a phase III trial in metastatic HER2-negative BC with germline BRCA1/2-mutation (see ClinicalTrials.gov Identifier: NCT02000622) [66,67]. In January 2018, it was licensed by the U.S Food and Drug Administration (FDA) for the treatment of patients with germline BRCA-mutated HER2-negative MBC who have previously received chemotherapy. Furthermore, *olaparib* demonstrated its efficacy in combination with Paclitaxel in a phase I/II randomized multicenter study in patients with Metastatic Triple-Negative Breast Cancer (MTNBC, see ClinicalTrials.gov Identifier: NCT00707707) [68].

Nowadays, different clinical studies are beginning to assess and confirm the use of *olaparib* in MBC (see ClinicalTrials.gov Identifiers: NCT03742245, NCT05629429, NCT05033756, NCT05340413, NCT03344965 [69]).

From another point of view, a randomized multicenter Phase II trial investigated the efficacy and safety of *dinaciclib* in patients with previously treated advanced BC (see ClinicalTrials.gov Identifier: NCT00732810) [70,71,72].

In summary, BC cells harbor defects in DNA double-strand break repair and, therefore, are hypersensitive to PARP inhibition, while CDK-1 is necessary in BRCA1-mediated S phase checkpoint activation, in cell proliferation, and is overexpressed in BC cells.

In this view, the combined inhibition of CDK1 and PARP-1 in BC treatment resulted in dramatically reduced cell growth [73].

In support of this, recently, Turdo et al. demonstrated that *olaparib* in combination with *dinaciclib* reduced the growth rate of Triple-Negative Breast Cancer (TNBC) BRCA mutated cells, sparing normal breast cells [74].

Considering that polytherapy presents several disadvantages (such as patient compliance reduction, risk of adverse drug interactions) over monotherapy, here we propose an in silico mixed ligand/structure-based design of the first-in-class CDK-1/PARP-1 dual inhibitors as anti-BC agents.

In particular, we report an innovative in silico hybrid and hierarchical virtual screening to identify new CDK-1 and PARP-1 dual target inhibitors. The use of an in-house ligand-based Biotarget Predictor Tool (BPT) allowed us to fast screen a large database of active molecules, which were further investigated through structure-based studies.

## 2. Results and Discussion

As depicted in Figure 2, the workflow of the in silico protocol proposed in this work consisted of both classical and innovative ligand and structure-based techniques. This hierarchical multistep procedure was applied to identify new promising dual-target modulators of CDK-1 and PARP-1 with a possible implication in the therapy of BC.

In detail, in the first phase of the protocol, the well-established molecular descriptor-based Biotarget Predictor Tool (BPT), developed by us and available online in the DRUDIT web-platform (DRUg DIscovery Tools, open access web service, www.drudit.com, accessed on 19 July 2023) [75], was applied. Subsequently, structure-based studies of molecular docking were integrated with a new in-house correlation approach to gain more insight into the binding mode and the mechanism of action of the selected hits [76,77]; as a last step, Molecular Dynamic Simulations (MDS) were conducted for the best ranked dual inhibitor on both target protein 3D structures.

In the next sections, the various steps of the protocol are described in detail.

### 2.1. Ligand-Based Studies

#### 2.1.1. Ligand-Based Template Building

The BPT tool applied in the first step of the in silico protocol is a ligand-based protocol capable of predicting the affinity of inputting small molecules against the desired target/s, virtualized in the DRUDIT web-platform by means of an appropriate process of molecular descriptor calculation and manipulation (for further detail see refs. [75,76,78,79]). Thus, by following the procedure described in the literature [75], a preliminary phase of ligand-based template building for the targets of interest (CDK-1 and PARP-1) was carried out, as reported below.

Two large databases of CDK-1 and PARP-1 known modulators were downloaded from the BindingDB [80], a reliable web-accessible source of experimentally determined protein-ligand binding affinities, where the *K_i_*, *K_d_*, IC_50_, EC_50_ values, and the corresponding target information for thousands of active molecules are available. In detail, a cut-off of activity IC_50_ < 100 nM was fixed to select the most active inhibitors (databases are accessible in Appendix A).

The two sets of inhibitors were then docked in the corresponding target X-ray structures, retrieved from the RCSB Protein Data Bank (RCSB PDB) [81,82] (PDB codes 6GU6 [83] and 7KK4 [84], for CDK-1 and PARP-1, respectively). The 3D best docked poses of each ligand “frozen” into the protein binding site were downloaded and submitted to a molecular descriptor calculation performed through our MOLDESTO software (MOlecular DEScriptors TOols) [75]. This yielded more than 1000 molecular descriptors (3D, 2D, and 1D) for each of the input structures. This preliminary docking allowed us to perform a finer calculation of 3D molecular descriptors, which were calculated for the best tridimensional orientation of the ligand into the protein binding site.

The resulting two compounds vs. molecular descriptor matrices (Appendix A) were converted into two sequences of value couples for each molecular descriptor (mean and standard deviation) which constituted the two molecular descriptor-based target templates [75].

#### 2.1.2. Biotarget Predictor Tool—Multitarget Mode

Once the target templates were built and integrated into the DRUDIT servers, the first ligand-based phase of the proposed protocol was focused on the virtual screening of a large structure database of small molecules. In this study, the National Cancer Institute (NCI) database, including about 38,910 compounds analyzed by the National Cancer Institute in in vitro antiproliferative assays against 60 cancer cell lines (NCI60) [85,86], was selected.

In details, the compounds were uploaded to the DRUDIT web service and processed with the BPT tool (default parameters were used, as reported in [87]), whose output matrix reported the predicted affinity of input structures weighted by the Drudit Affinity Score values (DAS, a parameter ranging in the range 0/1, low/high affinity) for the selected biological targets (Appendix A). Furthermore, as the main aim of the study was the identification of new dual CDK-1/PARP-1, the multitarget mode was applied by computing the “multitarget score” (MScore) parameter, defined by the equation:Multitarget Score = DAS_CDK-1_ × DAS_PARP-1_
(1)
where DAS_CDK-1_ and DAS_PARP-1_ represent the DAS score for CDK-1 and PARP-1 molecular descriptor-based templates, respectively. The multitarget score allowed for selecting structures with optimal activity against both targets: the higher the two DAS scores, the higher was the MScore, thus, the higher the probability for the small molecule to inhibit both targets.

The analyzed compounds were ranked according to this parameter and the MScore computed by applying Equation (1) to the DAS scores of the two reference compounds (DAS_CDK-1(*dinaciclib*)_ × DAS_PARP-1(*olaparib*)_) was selected as threshold value (0.737872): the 290 best ranked structures (Appendix A) were thus selected to conduct further in silico investigations.

### 2.2. Structure-Based Studies: Molecular Docking Analysis

In the second phase of the protocol, we evaluated, by means of two sequential molecular docking studies, the effective capability of the selected small molecules to insert deeply into the protein’s binding pockets and their ability to interact with key amino acids of the active sites.

A brief description of the selected targets and their binding sites is reported below.

From a structure point of view, inactive monomeric CDK-1 adopts a classical bi-lobal protein kinase fold with a smaller N-terminal lobe linked through a hinge to a larger C-terminal fold [88]. Inactivity depends on the inappropriate disposition of the activation segment (residues 146–173), the P loop (Gly-rich phosphate binding sequence, residues 11–17), and the C-helix (residues 47–57) [83]. Figure 3a shows the 3D X-ray structure of the inactive monomer of CDK-1 (PDB code 6GU6 [83]).

The ATP binding site (Figure 3b) is composed of an adenine pocket, characterized by a hydrogen bond recognition motif and a hydrophobic portion occupied by the nucleotide base. Regarding the nucleoside hydrogen bond recognition site, the backbone N-H of Leu^83^ forms hydrogen bonds with the adenine and the Glu^81^ backbone carbonyl with the substituted amine. The triphosphate portion of ATP binds among residues 142–148 and 30–36. Critical to the hydrolytic functionality of this region are residues Asp^145^ and Lys^33^ [89,90].

Of particular interest is the role of Phe^80^, which may function as a hydrophobic gate enabling the entering of the catalytic portion of the ATP site in a highly regulated manner.

Additionally, Thr^84^ is on the solvent accessible surface and its carbonyl is near the N-H and carbonyl of Leu^83^, forming, in the absence of ligands, favorable electrostatic attractions for the formation of a hydrogen-bonded network of solvents [83,89,90].

On the other hand, PARP-1 is structurally divided in three domains: the N-terminal DNA-binding domain (with three Zinc finger) [91], the central auto-modification domain (with specific glutamate and lysine residues as acceptors of ADP-ribose moieties) [92], and the C-terminal catalytic domain, which utilizes nicotinamide adenine dinucleotide (NAD^+^) as a substrate to construct polymers of ADP-ribose on histones. The catalytic domain is composed of two subdomains, the helical subdomain and the ART subdomain, essential for forming poly ADP-ribose polymer [93,94].

Figure 4a shows the PARP-1 catalytic domain, highlighting the main residues Gln^759^, Glu^763^, Asp^766^, Asn^767^, Gly^863^, Tyr^896^, Ala^898^, Ser^904^, and Tyr^907^ (highly responsible for the stability of the binding pocket), Gly^863^ and Ser^904^ (which form a hydrogen bond network with the nicotinamide moiety), Glu^988^ (catalytically important residue), Asp^770^ and Arg^878^ (critical in stabilizing the adenosine portion of the substrate NAD^+^), and Tyr^907^ (which forms a planar surface) [95,96,97] (Figure 4b).

As described in detail below, the first phase of the structure-based studies consisted of a Docking Virtual Screening Workflow (DVSW), a protocol available in the Maestro suite, including three subphases of semi-flexible docking analysis with increasing levels of accuracy.

In the second step of the molecular docking studies, an Induced Fit Docking (IFD) was performed for the best ranked compounds emerged from XP docking to evaluate, with an even higher reliability and accuracy, their capability to fit into the target binding sites.

#### 2.2.1. Docking Virtual Screening Workflow (DVSW)

In the first structure-based phase, DVSW from the Maestro suite was applied. The protocol, as described in the Material and Methods (Section 3), includes three consecutive steps: High-Throughput Virtual Screening (HTVS), Standard Precision (SP) docking, and Extra Precision (XP) docking, from less to the more accurate. The 290 selected molecules were filtered by keeping 50% of the best docked compounds at the end of each step.

Thus, docking grids were centered on the PARP-1 and CDK-1 binding pockets, including all the key amino acid residues. Figure 5a,b shows the 3D binding sites of PARP-1 in complex with *olaparib* (PDB code 7KK4 [84]) and CDK-1 in complex with *dinaciclib* (PDB code 6GU6 [83]), respectively.

The last of the three consecutive DVSW steps, the XP docking, kept the 35 best docked compounds for PARP-1 (PDB code 7KK4 [84]) and the 33 best docked compounds for CDK-1 (PDB code 6GU6 [83]), whose docking scores are given in Table 1. From the analysis of the docking studies results, it was possible to appreciate that eight compounds (**645656**, **670757**, **697678**, **711806**, **717843**, **732508**, **733301**, and **733303**, 2D structures in Figure 6) emerged as the best docked compounds against both PARP-1 and CDK-1.

Table 1 shows the docking scores of the filtered structures and the reference ligands *olaparib* and *dinaciclib*, highlighting compounds presented in both rankings.

These interesting results prompted us to further evaluate the binding mode of the eight compounds into the target protein catalytic sites by means of Induced Fit Docking (IFD) studies.

#### 2.2.2. Induced Fit Docking (IFD)

Induced Fit Docking (IFD) studies were performed on both PARP-1 (PDB code 7KK4 [84]) and CDK-1 (PDB code 6GU6 [83]).

Table 2 shows the IFD scores of the selected eight structures and the reference ligands *dinaciclib* and *olaparib*.

By analyzing the obtained IFD score range, we confirmed the capability of all of them to efficaciously interact with both targets, with IFD scores comparable to the reference ligands (Table 2).

Furthermore, this ability was confirmed by a detailed analysis of the key interactions formed by each compound with both protein binding sites.

In this view, Table 3 and Table 4 provide an overview of the amino acids, located at 4 Å, involved in the binding with compounds **645656**, **670757**, **697678**, **711806**, **717843**, **732508**, **733301**, **733303**, and the two reference compounds both on PARP-1 and CDK-1, respectively.

As shown in Table 3, all the selected derivatives formed a total number of interactions comparable to the already approved PARP-1 inhibitor *olaparib* (in the range of 20–24 vs. 24 interactions for *olaparib*).

In detail, all compounds were involved in stabilizing interactions with key residues of the binding pocket (Tyr^689^, Glu^763^, Asp^766^, Asn^767^, Leu^769^, Trp^861^, His^862^, Gly^863^, Ser^864^, Arg^865^, Asn^868^, Ile^872^, Gly^876^, Leu^877^, Ile^879^, Ala^880^, Pro^881^, Tyr^889^, Met^890^, Phe^891^, Gly^892^, Lys^893^, Gly^894^, Ile^895^, Tyr^896^, Phe^897^, Ala^898^, Lys^903^, Ser^904^, Tyr^907^, His^909^, Leu^984^, and Asn^987^), but also with the catalytically important residue Glu^988^ and residues critically involved in stabilizing the adenosine portion of the substrate NAD^+^, Asp^770^ and Arg^878^.

Essential requirements, presented by highly active known inhibitors of PARP-1, are met by the selected compounds: Pi-Pi stacking interactions with Tyr^907^ were formed by six of the eight compounds, among which **645656** and **711806** stabilized the pose by means of a double Pi-Pi stacking bonds network; Tyr^689^, Asp^766^, Trp^861^, His^862^, Gly^863^, Ser^864^, Arg^878^, Ala^880^, Tyr^889^, Gly^894^, Ser^904^, and Glu^988^ act as H-bond donors or acceptors for all compounds. Particularly recurrent are H-bonds with a Gly^863^ backbone and Ser^904^ side chain; hydrophobic interactions with Ala^898^ and Glu^988^ are periodic in all compounds.

Table 4 denotes that the eight compounds formed a total number of interactions comparable to the already approved CDK-1 inhibitor *dinaciclib* (in the range of 19–24 vs. 23 interactions for *dinaciclib*).

All compounds can interact with CDK-1 ATP binding site crucial amino acids such as Ile^10^, Gly^11^, Glu^12^, Gly^13^, Val^18^, Ala^31^, Lys^33^, Val^64^, Phe^80^, Glu^81^, Phe^82^, Leu^83^, Thr^84^, Met^85^, Asp^86^, Lys^89^, Glu132, Leu^135^, and Asp^145^. Among the listed residues, Asp^145^ and Lys^33^ are critical to the hydrolytic functionality, Leu^83^ and Glu^81^ are essential in the H-bond’s formation with the adenine and can stabilize the selected compounds through 1/2 H-bonds, Phe^80^ acts as a hydrophobic gate, Ile^10^, Gly^11^, Glu^12^, Gly^13^, Val^18^, Ala^31^, Val^64^, Phe^82^, Met^85^, Asp^86^, Lys^89^, Glu^132^, and Leu^135^ are involved in the connection between the surface of the ATP binding domain and the cyclin binding domain, and Thr^84^ is essential in the formation of favorable electrostatic attraction. Compounds **645656**, **670757**, **711806**, **717843**, **732508**, and **733303** can stabilize the pose through an additional H-bond with Asp^146^, as for *dinaciclib*, and **645656** forms also extra interactions with Tyr^15^ and Lys^88^, while Thr^14^, Lys^20^, Lys^130^, and Asn^133^ can interact with few of the eight compounds.

### 2.3. Correlation Analysis between Antiproliferative Activity and CDK-1/PARP-1 Expression across NCI60 Panel

To further investigate the putative dual-target mechanism of action of the selected compounds, an additional correlation analysis between drug activity and protein expression data, which we recently proposed and applied [76,77], was conducted.

Supposing that the correlation between the antiproliferative activity data and the target protein’s expression pattern could allow us to presume a mechanism of action for input structures, we collected experimental data for the screened ligands and the targets of interest from the NCI database.

The NCI60 Human Tumor Cell Lines Screen is a best-known project based on the large-scale screening of chemical compounds and cancer cell phenotypes, consisting of standardized assays performed on approximately sixty cancer cell lines belonging to nine different subpanels (leukemia, non-small-cell lung, colon, central nervous system, melanoma, ovarian, renal, prostate, and breast cancer cells) characterized at genomic, transcriptomic, and proteomic levels.

In detail, we downloaded three different data sets: antiproliferative activity data (expressed as GI_50_ values) of the eight ligands and expression pattern data for 60 different human tumor cell lines for both CDK-1 and PARP-1 that includes 16 and 8 experiments, respectively. The rationale behind this approach is that if the antiproliferative activity of a molecule is well linked to the protein expression pattern (high/low antiproliferative effect in a cancer cell line with high/low expression of target protein), it is likely that the modulation of that target is responsible for the ligand antiproliferative activity.

#### Data Normalization and Matching

Collected NCI data, derived from different types of experiments, are heterogeneous and characterized by various measurement units. Therefore, a normalization process was necessary, for both the antiproliferative activity and protein expression pattern data, to make them homogeneous and comparable.

In detail, we obtained the molecular target expression pattern values (EP_i_) and the antiproliferative activity values (GI_50,i_) against the 60 tumoral cell lines and their mean values (µ_P_ and µ_G_, respectively). The deviation of each EP_i_ from µ_P_ and of each GI_50,i_ from µ_G_, normalized against the highest absolute value (M-GI_50,i_ and M-EP_i_), was computed to obtain Normalized Expression Pattern values (NEP_i_) and Normalized Antiproliferative activity values (NGI_50,i_).

Finally, normalized data were matched according with Formula (2) (δ_i_) and a fitting score (Φ) was computed for both CDK-1 and PARP-1 by applying the Formula (3):δi = NEP_i_ × NGI_50,i_(2)
Φ = Σδ_i_(3)

Figure 7 explains the workflow for the assessment of the correlation between the antiproliferative activity values and the expression patterns, from the normalization data process to the data normalized matching.

A matching score value φ (0/60, lowest/highest values), which expressed the correlation between protein expression pattern and chemosensitivity, was assigned to each structure.

Normalized data are available in Appendix A.

For each structure, we took into consideration the maximum matching value for both CDK-1 and PARP-1; then, the mean value was computed (Table 5).

Among all, compound **645656** showed the highest Mean Matching score value. By analyzing its 2D structure, a benzimidazole moiety is found to constitute the small molecule central core, which represents an attractive structural class due to its relatively low molecular weight and high intrinsic potency [98,99]. Thus, compound **645656** was selected as the most interesting compound to be analyzed with Molecular Dynamic Simulations.

### 2.4. Molecular Dynamic Simulations

Molecular Dynamic Simulations were performed to gain insight into the structural features of **645656**/CDK-1 (PDB code 6GU6 [83]) and **645656**/PARP-1 (PDB code 7KK4 [84]) complexes, analyzing the mutual conformational changes between **645656** and proteins in a 100 ns timescale.

As shown in Figure 8, Induced Fit Docking studies show that compound **645656** can adopt itself into both PARP-1 and CDK-1 binding sites, interacting with key residues in each case. In detail, the benzimidazole ring could interact, through its nitrogen atoms, with the side chain of His^862^ and the backbone of Asp^766^, forming two hydrogen bonds, while naphthalene moiety stabilizes the ligand by two Pi-Pi staking interactions with the phenyl ring of Tyr^907^ at the PARP-1 catalytic site (Figure 8a). Simultaneously, the two nitrogen benzimidazole atoms form two hydrogen bonds with the -NH_2_ side chain of Gln^132^ and the backbone of Asp^86^, while the hydroxydril group of **645656** interacts with Asp^146^ through a third hydrogen bond at the CDK-1 binding site (Figure 8b).

The MDS trajectories provide key information on the stability and relationship of various molecular interactions on the complexes through their Root Mean Square Deviation (RMSD).

The RMSD was calculated for the simulation trajectory of 100 ns for the ligand and protein. It was intended to measure the average change in the displacement of the backbone.

Compound **645656** achieved an acceptable stability inside both the binding sites, confirming a potential dual-target inhibition activity. Indeed, as depicted in Figure 9a,b (variation in the ligand and CDK-1/PARP-1 RMSDs across the first 20 ns simulation time), the RMSD values for both proteins and ligand are within the acceptable range of 1-3Å, confirming the stability of the complex. RMSDs for reference complexes, *olaparib*/PARP-1 and *dinaciclib*/CDK-1, have been also calculated and are available in Appendix A.

MDS studies also depict plots of protein–ligand contacts and explains the interaction fraction of the protein residue with the ligand, which explains how much (%) of the simulation time of the specific interaction is maintained between ligand and receptors complexes. Figure 10a,c shows protein–ligand contacts for **645656**/PARP-1 and **645656**/CDK-1, respectively. Concerning the **645656**/PARP-1 complex, Glu^763^, Asn^766^, His^862^, Arg^878^, Tyr^896^, and Tyr^907^ showed the highest interaction fractions, among which Arg^878^ ranged from 1 to 1.2. On the other hand, the **645656**/CDK-1 complex preserved good interaction fractions with Ala^31^, Lys^33^, Asp^86^, Gln^132^, Leu^135^, and Ala^145^.

Furthermore, Figure 10b,d reports a detailed schematic diagram of protein–ligand interactions that occur more than 30% of the MD simulation time.

The response was further studied in terms of protein and ligand binding energy, demonstrating a high stability across the simulation time and reaching a plateau (time-energy graphs for both complexes are depicted in Figure 11a,b). Energy vs. Time plots were investigated also for reference ligands, which reached a plateau after 30 ns of simulations (time-energy graphs for both reference complexes, *olaparib*/PARP-1 and *dinaciclib*/CDK-1, are depicted in Appendix A).

Protein Root Main Square Fluctuation (RMSF, Appendix A), Ligand RMSF (Appendix A), Ligand RMSD (Appendix A), Radius of Gyration (rGyr, Appendix A), and Binding Free Energy (Appendix A) have been computed for all four complexes (**645656**/CDK-1, **645656**/PARP-1, *olaparib*/PARP-1, and *dinaciclib*/CDK-1).

## 3. Materials and Methods

### 3.1. Ligand-Based Studies

The web service DRUDIT (www.drudit.com, accessed on 19 July 2023) operates on four servers, each of which can perform more than ten jobs simultaneously, running several software modules implemented in C and JAVA on MacOS Mojave. The Biotarget Finder Module was used in a multitarget mode to screen the large NCI database of active small molecules as CDK-1 and PARP-1 inhibitors in breast cancer treatment [75].

#### Biotarget Predictor Tool (BPT)

The tool provides prediction of the binding affinity between candidate molecules and the specified biological target. Templates of CDK-1 and PARP-1 were created using two sets of well-known protein inhibitors. Molecular docking studies were performed at both the CDK-1 and PARP-1 binding sites to freeze ligands into the pockets. Molecular descriptors were calculated through MOLDESTO. The two built molecular descriptor target templates were implemented in DRUDIT and the default DRUDIT parameters (*N* = 500, *Z* = 50, *G* = a) were used [75,87]. In accordance with the first phase of the in silico workflow, the NCI database was uploaded to DRUDIT and submitted to the Biotarget Predictor in a multitarget mode. The output results were obtained as a DAS (Drudit Affinity Score) value for each structure, reflecting the binding affinity of compounds against both CDK-1 (DAS_CDK-1_) and PARP-1 (DAS_PARP-1_). 

### 3.2. Structure-Based Studies

The preparation process of ligands and proteins used for in silico studies was performed as detailed below:

#### 3.2.1. Ligand Preparation

The ligands for docking were prepared using the LigPrep tool from the Schrödinger Maestro Suite [100]. For each ligand, all possible tautomers and stereoisomers were generated for a pH of 7.0 ± 0.4, using default setting, through the Epik ionization method [101]. Consequently, the integrated Optimized Potentials for Liquid Simulations (OPLS 2005) force field was used to minimize the energy status of the ligands [102].

#### 3.2.2. Protein Preparations

The crystal structures of CDK-1 and PARP-1 (PDB codes 6GU6 [83], 7KK4 [84], respectively,) were downloaded from the Protein Data Bank [81,82] and prepared using the Protein Preparation Wizard in the Schrödinger software with default settings [103]. In detail, bond orders, including the Het group, were assigned hydrogen atoms all water molecules were deleted, and protonation of the heteroatom states was carried out using the Epik-tool (with the pH set at biologically relevant values, i.e., at 7.0 ± 0.4). The H-bond network was then optimized. The structures were finally subjected to a restrained energy minimization step (RMSD of the atom displacement for terminating the minimization was 0.3 Å) using the OPLS 2005 force field [102].

#### 3.2.3. Docking Validation

Molecular Docking studies were executed and scored using the Glide module from the Schrödinger Suite. The receptor grids were obtained through assignment of the original ligands (*dinaciclib* and *olaparib* for PDB codes 6GU6 [83] and 7KK4 [84], respectively) as the centroid of the grid boxes. Extra Precision (XP) mode, as scoring function, was used to dock the generated 3D conformers into the receptor model. The post-docking minimization step was performed with a total of 5 poses for each ligand conformer and a maximum of 2 docking poses were generated per ligand conformer. The proposed docking procedure was able to re-dock the original ligands within the receptor-binding pockets with RMSD  < 0.51 Å.

#### 3.2.4. Docking Virtual Screening Workflow (DVSW)

The Virtual Screening Workflow was used to screen the DRUDIT-selected compounds against CDK-1 and PARP-1 (PDB codes 6GU6 [83] and 7KK4 [84], respectively). The full workflow includes Glide docking at the three accuracy levels. The first stage performed HTVS docking and 50% of ligands were retained to pass to the next stage, which performed SP docking. Again, 50% of the survivors of this stage were passed on to the third stage, which performed XP docking.

#### 3.2.5. Induced Fit Docking

Induced Fit Docking simulation was performed using the Induced Fit Docking (IFD) application, an accurate and robust Schrödinger technology that accounts for both ligand and receptor flexibility [104,105]. Schrödinger’s IFD-validated protocol was applied using CDK-1 and PARP-1 proteins from the PDB (PDB codes 6GU6 [83] and 7KK4 [84], respectively), previously refined by the Protein Preparation module. The IFD score (IFD score = 1.0 Glide Gscore + 0.05 Prime Energy), which includes protein–ligand interaction energy and system total energy, was calculated and used to rank the IFD poses.

#### 3.2.6. Molecular Dynamic Simulation

Molecular Dynamics Simulations were performed using Desmond software to confirm the binding stability and strength of **645656**/CDK-1 (PDB code 6GU6 [83]) and **645656**/PARP-1 (PDB code 7KK4 [84]) complexes. The constant-temperature–constant-pressure ensemble (NPT) allowed to control both temperature and pressure. The unit cell vectors are allowed to change and pressure is controlled by adjusting volume. System Temperature and Pressure were set at 300 K and 1.01325 bar, respectively, Systems energy was minimized for 1000 steps before a production run of 20 ns for both complexes. The results were analyzed in terms of protein and ligand time-lapse binding energy, RMSD, and protein–ligand contact.

## 4. Conclusions

In accordance with their crucial role in the cell cycle and DNA repair and their overexpression in cancer cells, CDK-1 and PARP-1 were demonstrated to be key targets in BC progression. Compounds acting as CDK-1 and/or PARP-1 inhibitors are effective in causing cell death in BC but not in normal cells through a selective synthetic lethality mechanism; the treatment with mixed therapy by means of CDK-1 and PARP-1 inhibitors resulted in a radical cell growth reduction.

However, the advantages/disadvantages balance is greater for monotherapy, which allows for an improvement in patient compliance, a reduction in the risk of adverse drug interactions, and easier identification of the desired therapeutic effect and possible undesirable reactions. In this view, here we proposed an in silico mixed ligand/structure-based design of the first-in-class CDK-1/PARP-1 dual inhibitors as anti-BC agents.

In detail, we identified through a combined structure and ligand-based virtual screening compound **645656**—already known in the literature for its antibacterial activity [106]—as a potential CDK-1 and PARP-1 inhibitor, acting with a dual-target inhibition mechanism, as a potential effective monotherapy in BC.

The Biotarget Predictor Tool was used to screen the large NCI database in a multitarget mode to select only compounds with the best simultaneous affinity against both CDK-1 and PARP-1. The 290 best-scored structures were further analyzed with hierarchical docking studies consisted of High-Throughput Virtual Screening (HTVS), Standard Precision (SP) docking, Extra Precision (XP) docking, and Induced Fit Docking (IFD), allowing us to screen the eight best ranked ligands for successive analysis.

In particular, the putative dual-target mechanism of action was investigated through the correlation between the antiproliferative activity data and the expression pattern of the target proteins (CDK-1 and PARP-1). Among all the eight screened structures, compound **645656** showed the highest matching value. Therefore, we decided to perform a further structure-based analysis by means of Molecular Dynamic Simulation.

The Root Mean Square Deviation (RMSD), protein–ligand interaction fractions, and protein–ligand binding energy examination demonstrated the high stability of both **645656**/PARP-1 and **645656**/CDK-1 complexes, confirming the ability of compound **645656** to interact with the target’s binding sites with a potential dual-target inhibition mode of action.

## Figures and Tables

**Figure 1 ijms-24-13769-f001:**
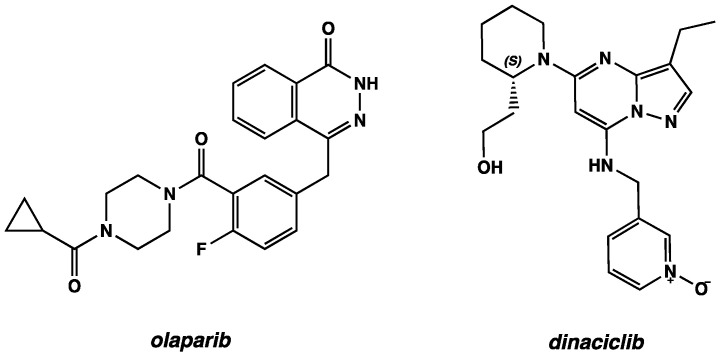
2D chemical structure of *olaparib* and *dinaciclib*.

**Figure 2 ijms-24-13769-f002:**
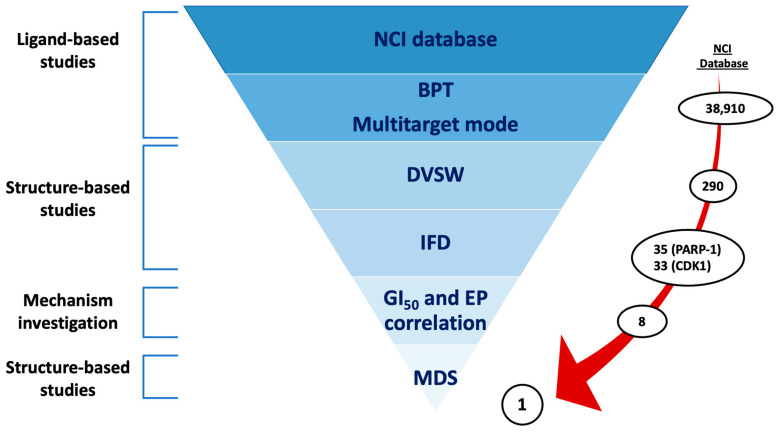
In silico hybrid protocol flowchart for the identification of new dual-target strategy in the treatment of breast cancer.

**Figure 3 ijms-24-13769-f003:**
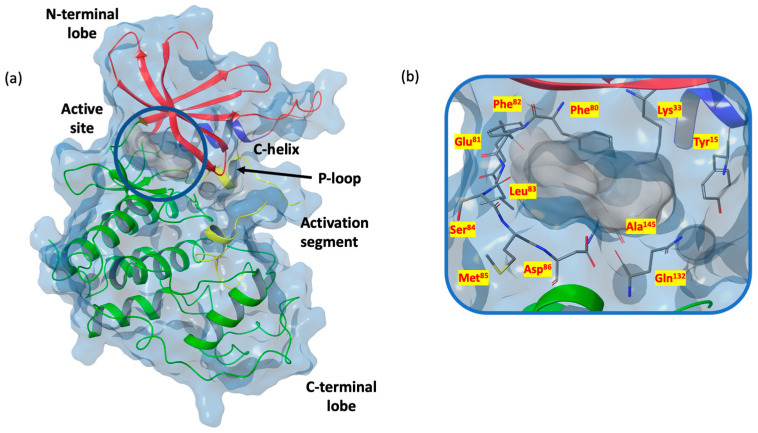
(**a**) X-ray structure of the inactive monomer of CDK-1 (PDB code 6GU6); N- and C-terminal lobes are labeled in red and green, respectively, key regulatory elements C-helix and activation segment are in blue and yellow; (**b**) 3D ATP binding site of CDK-1 (PDB code 6GU6).

**Figure 4 ijms-24-13769-f004:**
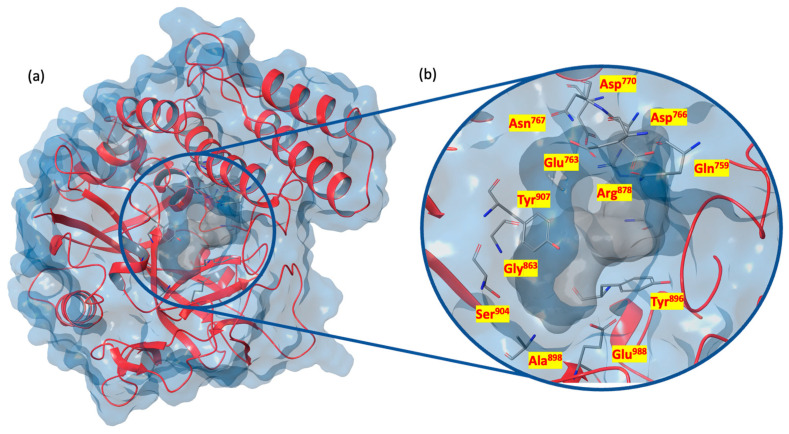
(**a**) X-ray structure of PARP-1 C-terminal catalytic domain (PDB code 7KK4); (**b**) focus on the binding pocket, with the main residues highlighted (PDB code 7KK4).

**Figure 5 ijms-24-13769-f005:**
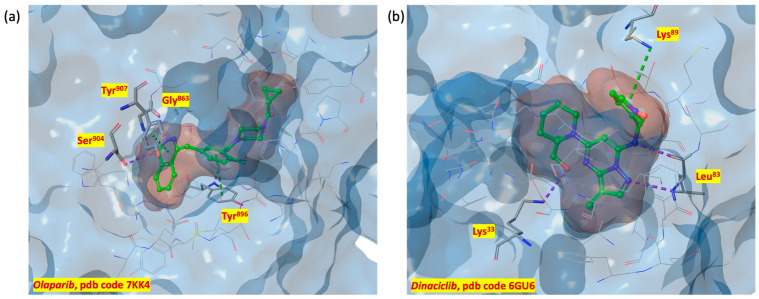
(**a**) PARP-1 3D binding surface in complex with *olaparib* (PDB code 7KK4); (**b**) CDK-1 3D binding surface in complex with *dinaciclib* (PDB code 6GU6).

**Figure 6 ijms-24-13769-f006:**
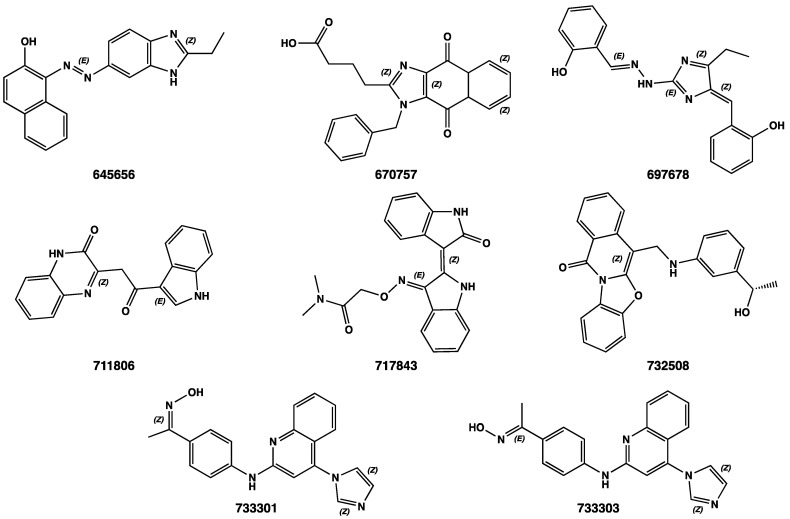
2D chemical structures of compounds 645656, 670757, 697678, 711806, 717843, 732508, 733301, and 733303.

**Figure 7 ijms-24-13769-f007:**
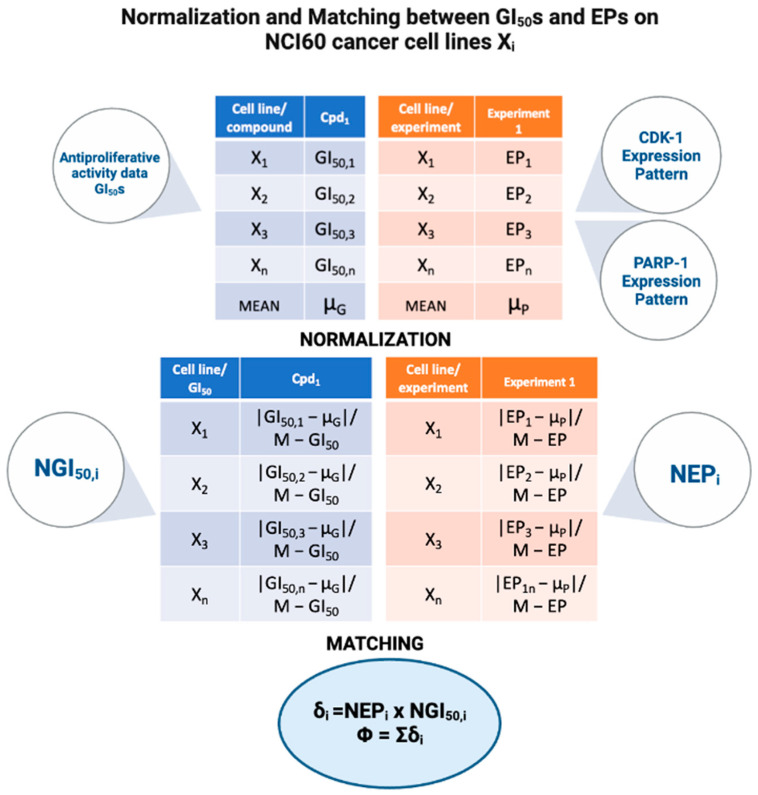
Workflow for the assessment of the correlation between the antiproliferative activity values (expressed as GI_50_s) and expression patterns (Eps) on NCI60 cancer cell lines χ_i_.

**Figure 8 ijms-24-13769-f008:**
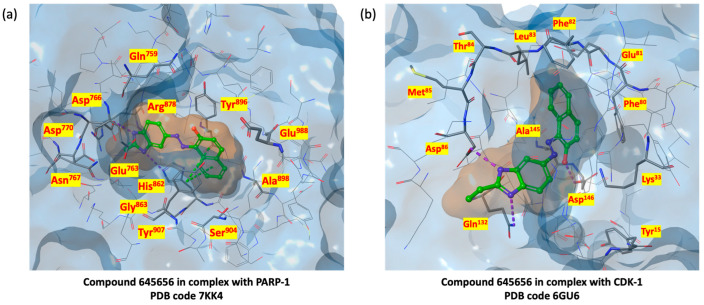
(**a**) PARP-1 3D catalytic site surface in complex with **645656** (PDB code 7KK4); (**b**) CDK-1 3D binding site surface in complex with **645656** (PDB code 6GU6).

**Figure 9 ijms-24-13769-f009:**
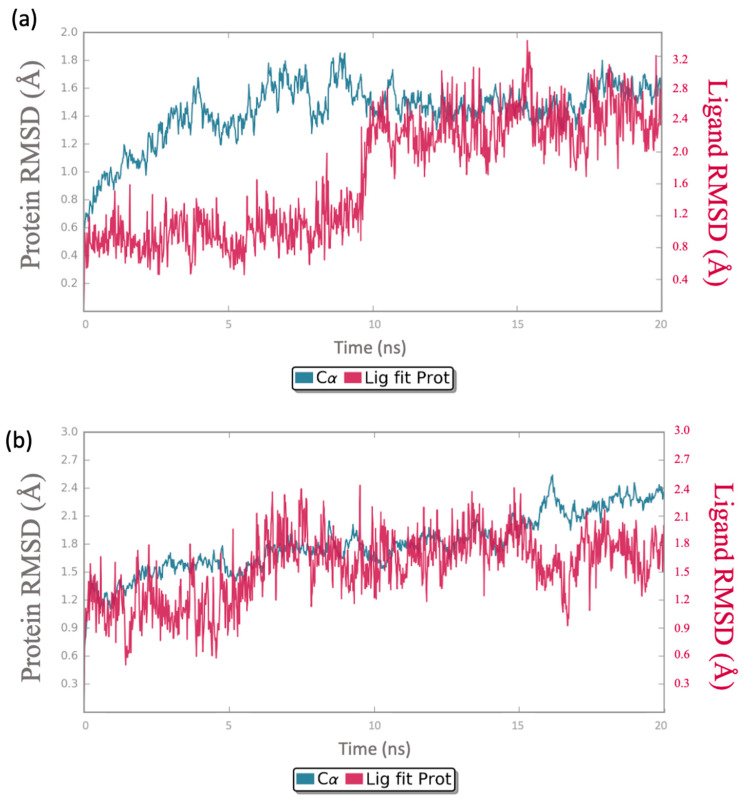
(**a**) Calculated RMSD during the first 20 ns of the simulation trajectory for **645656**/PARP-1 complex; (**b**) calculated RMSD during the first 20 ns of the simulation trajectory for **645656**/CDK-1 complex.

**Figure 10 ijms-24-13769-f010:**
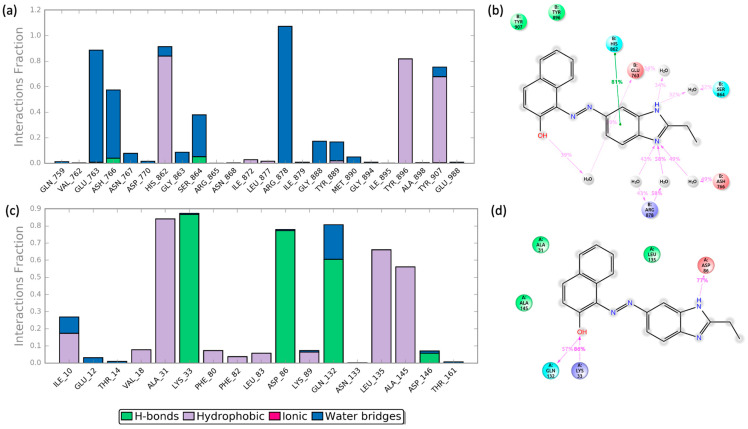
(**a**) Protein–ligand interactions examination across the simulation time for **645656**/PARP-1 complex; (**b**) schematic of detailed ligand atom interactions with PARP-1 residues; hydrogen bonds are labeled in violet, while Pi-Pi stackings are in green; (**c**) protein-ligand interactions examination across the simulation time for **645656**/CDK-1 complex; (**d**) schematic of detailed ligand atom interactions with CDK-1 residues; hydrogen bonds are labeled in violet.

**Figure 11 ijms-24-13769-f011:**
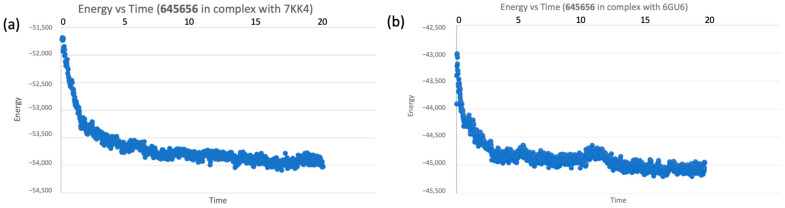
(**a**) Molecular dynamics simulation for compound **645656** in complex with 7KK4, graph of energy variation during the first 30 ns of the simulation time; (**b**) molecular dynamics simulation for compound **645656** in complex with 6GU6, graph of energy variation during the first 20 ns of the simulation time.

**Table 1 ijms-24-13769-t001:** Docking scores of the structures that emerged from the DVSW; compounds **645656**, **670757**, **697678**, **711806**, **717843**, **732508**, **733301**, and **733303**, kept in both the DVSWs, are labeled in bold.

PARP-1		CDK-1	
CPD	Docking Score	CPD	Docking Score
*olaparib*	−13.933	717838	−10.697
694470	−11.395	*dinaciclib*	−9.832
697767	−10.552	**670757**	−9.095
694113	−10.323	**717843**	−8.676
724117	−10.197	720565	−8.666
744227	−10.02	699250	−8.660
299589	−9.618	766478	−8.573
694962	−9.395	**733301**	−8.463
690659	−9.263	642635	−8.455
673321	−8.778	**733303**	−8.395
**711806**	−8.749	681700	−8.176
701592	−8.681	**732508**	−8.093
761910	−8.632	**697678**	−8.008
711066	−8.485	646922	−7.865
**697678**	−8.424	**711806**	−7.796
692427	−8.323	653022	−7.557
688537	−8.305	645814	−7.525
**733301**	−8.272	710858	−7.448
717309	−8.223	655350	−7.425
636785	−8.189	670532	−7.415
697763	−8.170	706028	−7.311
**717843**	−8.118	**645656**	−7.300
707442	−8.056	766294	−7.254
169874	−8.035	652810	−7.037
760217	−8.009	655349	−6.990
**645656**	−7.985	692634	−6.886
665325	−7.929	732491	−6.870
668266	−7.856	711803	−6.811
**670757**	−7.821	657350	−6.805
705935	−7.777	665314	−6.551
**733303**	−7.743	677945	−6.525
724350	−7.737	645614	−6.466
665712	−7.722	699238	−6.389
745813	−7.716	751166	−6.342
**732508**	−7.691	*-*	-
654632	−7.685	*-*	-

**Table 2 ijms-24-13769-t002:** IFD scores of the eight selected structures against both CDK-1 (PDB code 6GU6 [83]) and PARP-1 (PDB code 7KK4 [84]).

CDK-1 (PDB Code 6GU6)		PARP-1 (PDB Code 7KK4)	
Title	IFD Score	Title	IFD Score
*dinaciclib*	−620.868	*olaparib*	−751.917
**733301**	−619.486	**733303**	−750.076
**733303**	−618.721	**733301**	−748.649
**697678**	−617.850	**697678**	−748.335
**732508**	−617.331	**732508**	−748.010
**645656**	−617.083	**645656**	−747.542
**711806**	−616.539	**711806**	−746.817
**670757**	−616.136	**670757**	−746.191
**717843**	−614.408	**717843**	−744.721

**Table 3 ijms-24-13769-t003:** Overview of the amino acids involved in the binding of the selected compounds **645656**, **670757**, **697678**, **711806**, **717843**, **732508**, **733301**, **733303**, and *olaparib* in the binding site of PARP-1 at 4 Å proximity.

Title		*olaparib*	645656	670757	697678	711806	717843	732508	733301	733303
PARP-1 binding site	Y^689^			X *						
	E^763^						X	X		
	D^766^	X	X *	X	X	X	X	X	X	X *
	N^767^			X	X		X	X		
	L^769^	X								X
	D^770^	X	X		X	X			X	X
	W^861^	X	X	X			X	X	X *	X
	H^862^	X	X *	X	X	X ^§^ X ^§^ X ^§^	X	X	X	X
	G^863^	X * X *	X	X *	X *	X *	X * X *	X *	X	X *
	S^864^		X	X	X *	X	X	X	X	
	R^865^			X ^§^			X	X		
	N^868^		X	X	X	X	X			
	I^872^		X		X	X			X	
	G^876^		X							
	L^877^		X			X			X	
	R^878^	X	X		X *	X *		X	X	X
	I^879^	X			X	X		X	X	X
	A^880^	X	X	X	X			X *	X *	X
	P^881^	X								X
	Y^889^	X *	X	X	X		X	X ^§^	X ^§^	X
	M^890^	X					X			X
	F^891^						X			
	G^892^	X			X			X	X	
	K^893^				X			X	X	X
	G^894^	X		X	X	X		X *	X	X
	I^895^	X				X		X	X	
	Y^896^	X * X ^§^	X	X	X	X	X	X	X	X ^§^
	F^897^	X	X	X	X	X	X	X	X	X
	A^898^	X	X	X	X	X	X	X	X	X
	K^903^	X	X	X	X	X	X ^§^	X	X	X
	S^904^	X *	X	X *	X *	X	X *	X	X *	X *
	Y^907^	X ^§^	X ^§^ X ^§^	X ^§^	X ^§^	X ^§^ X ^§^	X *	X ^§^	X ^§^	X
	H^909^			X			X	X		
	L^984^									
	N^987^						X			
	E^988^	X	X	X	X	X	X	X	X	X
	Tot.	24	21	20	22	22	22	24	23	21

* H-bonds; ^§^ Pi-Pi stacking.

**Table 4 ijms-24-13769-t004:** Overview of the amino acids involved in the binding of the selected compounds **645656**, **670757**, **697678**, **711806**, **717843**, **732508**, **733301**, **733303**, and *dinaciclib* in the binding site of CDK-1 at 4 Å proximity.

Title		*dinaciclib*	645656	670757	697678	711806	717843	732508	733301	733303
CDK-1 binding site	I^10^	X	X	X	X	X	X	X	X	X
	G^11^	X	X	X	X	X	X	X	X	X
	E^12^	X	X	X	X	X	X	X	X	X *
	G^13^	X	X	X	X	X	X	X	X	X
	T^14^	X *				X	X	X		
	Y^15^		X					X		
	V^18^	X	X	X	X	X	X	X	X	X
	K^20^									X *
	A^31^	X	X	X	X	X	X	X	X	X
	K^33^	X	X	X	X	X	X *	X *	X ^§^	X ^§^
	V^64^	X	X	X	X	X	X	X	X	X
	F^80^	X	X	X	X	X ^§^	X	X	X	X
	E^81^	X	X	X	X *	X	X *	X	X	X
	F^82^	X	X	X	X	X	X	X	X	X
	L^83^	X * X *	X	X *	X * X *	X *	X * X *	X *	X * X *	X * X *
	T^84^	X	X	X	X *	X	X	X	X	X
	M^85^	X	X	X	X	X	X	X	X	X
	D^86^	X ^#^	X *	X	X	X	X	X	X	X
	K^88^		X							
	K^89^	X ^#^	X	X * X ^§^	X	X	X	X	X *	X
	K^130^						X	X		
	E^132^	X	X *	X	X	X	X	X *	X	X *
	N^133^	X					X			X
	L^135^	X	X	X	X	X	X	X	X	X
	A^145^	X	X	X	X	X	X	X	X	X
	D^146^	X *	X *	X		X *	X	X *		X
	Tot.	23	22	21	20	21	24	23	20	23

* H-bonds; ^#^ Salt bridges; ^§^ Pi-Pi stacking.

**Table 5 ijms-24-13769-t005:** Collection of the correlation between protein expression pattern (EPs) and antiproliferative activity (GI50s) data both for CDK-1 and PARP-1 and mean values for the eight small molecules selected as potential dual target (CDK-1 and PARP-1) modulators.

PDB Code	6GU6 (CDK-1)	7KK4 (PARP-1)	Mean Matching
Title	Matching	Matching	
**645656**	39	41	40
**697678**	39	30	35
**733301**	37	33	35
**711806**	34	33	34
**717843**	35	33	34
**732508**	32	31	32
**733303**	34	25	30
**670757**	25	28	27
*dinaciclib*	36	-	-
*Olaparib*	-	40	-

## Data Availability

Not applicable.

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
