# Peer review of "In Silico Mixed Ligand/Structure-Based Design of New CDK-1/PARP-1 Dual Inhibitors as Anti-Breast Cancer Agents"

_ijms, 2023, doi:10.3390/ijms241813769_

Round 1
Reviewer 1 Report
I can state that the topic is very interesting and results of the study could be helpful for to readers in the area of medical and pharmaceutical sciences. However, the IJMS journal is famous for its high-quality and comprehensive articles with high citation score. In reviewed manuscript selected paragraphs require some correction. I believe after some simple changes this paper is viable for publication and I will list my comment below. For this reason, I recommend publication after minor revision.
Detailed review
After reading the manuscript in detail, I conclude that the abstract, introduction, and other chapters cover the issues discussed in an extensive and proper manner. The conclusions presented by the authors are related to the main research issue.
The weakness of the reviewed work concern as follows:
- The small number of citations of works from the last 5 years is rather surprising (out of 91 bibliographic items, only 20 works have been published since 2018). Authors need to conduct detailed survey of the literature and supplement the citation with references to recent studies in the field.
- Moreover, some bibliographic records need to improve, e.g. reference no 31 is incomplete. Also, in reference no 67 the date of access is quite early. In this case, the protein model is actual, but in my opinion, before the publication In my opinion, the authors should check if a newer protein model is not available.
- Figure 6 needs to improve. Atom symbols in compunds no 697678, 711806, 732508, 733303 are in wrong orientation.
- What I am missing from the results is the indication of the value of the inhibition constant (Ki or pKi). Certainly, the authors obtained these values during the experiment. Including them at least in the supplementary file would be very useful for other researchers to compare their in vitro analyses.
- Individual typos in the text, e.g. in line 468 "delated" instead of "deleted"
I recommend publication after minor revision.
Author Response
We would like to thank you Reviewer 1 for the positive comments and the constructive analysis of our manuscript.
We have followed, point-by-point, your suggestions, and we hope that the revised version could be reconsidered for submission to the International Journal of Molecular Sciences.
You can find below the details of the revision steps:
I can state that the topic is very interesting, and results of the study could be helpful for to readers in the area of medical and pharmaceutical sciences. However, the IJMS journal is famous for its high-quality and comprehensive articles with high citation score. In reviewed manuscript selected paragraphs require some correction. I believe after some simple changes this paper is viable for publication and I will list my comment below. For this reason, I recommend publication after minor revision.
Detailed review
After reading the manuscript in detail, I conclude that the abstract, introduction, and other chapters cover the issues discussed in an extensive and proper manner. The conclusions presented by the authors are related to the main research issue.
The weakness of the reviewed work concern as follows:
- The small number of citations of works from the last 5 years is rather surprising (out of 91 bibliographic items, only 20 works have been published since 2018). Authors need to conduct detailed survey of the literature and supplement the citation with references to recent studies in the field.
Response: in order to improve and update citation with references, a detailed survey of the literature was carried out. In particular, in addition to the relevant articles already cited in the text, we added fifteen more recent articles going from 2018 to 2022.
- Moreover, some bibliographic records need to improve, e.g. reference no 31 is incomplete. Also, in reference no 67 the date of access is quite early. In this case, the protein model is actual, but in my opinion, before the publication. In my opinion, the authors should check if a newer protein model is not available.
Response: reference 31 has been completed (now it is reference 45). Moreover, the link to the PDB web page has been checked and the access date has been modified accordingly.
- Figure 6 needs to improve. Atom symbols in compounds no 697678, 711806, 732508, 733303 are in wrong orientation.
Response: figure 6 has been corrected.
- What I am missing from the results is the indication of the value of the inhibition constant (Ki or pKi). Certainly, the authors obtained these values during the experiment. Including them at least in the supplementary file would be very useful for other researchers to compare their in vitro analyses.
Response: We have computed the Binding Free Energy for olaparib/PARP-1, dinaciclib/CDK-1, 645656/PARP-1, and 645656/CDK-1 complexes, which represents the free energy change for the protein-inhibitor interaction (ΔG). This is used to determine the inhibition constant (ki) which is, in turn, the dissociation constant (Kd) of the protein-inhibitor complex. Therefore, we have calculated a parameter very close to the Ki or pKi. These results are available in Supplementary Material, Table S1.
- Individual typos in the text, e.g. in line 468 "delated" instead of "deleted"
Response: in line 468, “delated” has been changed with “deleted”, moreover all the text has been revised and other individual typos and grammatical mistakes have been corrected.
Reviewer 2 Report
The manuscript titled “In silico mixed ligand/structure-based design of new CDK- 1/PARP-1 dual inhibitors as anti-breast cancer agents” described two techniques, mixed ligand- and structure-based virtual screening which are used in identifying a novel compound known as 645656. Compound 645656 showed a significant affinity for both CDK-1 and PARP-1 enzymes which suggested its antiproliferative activity. Molecular dynamics simulations were carried out by the authors, to investigate the stability of the novel compounds within two proteins.
It is an interesting manuscript, the introduction is well-organized, the methods and results are well-represented, and the figures are clear and significant. My overall comment is accepting the manuscript after addressing some issues.
1- Authors carried out molecular dynamics simulations for 20ns, which is not enough time for the system to reach equilibrium. I suggested carrying out MD for 150ns at least to investigate the stability of the compound.
2- Authors need to carry out molecular dynamics simulation for reference compounds that are known by their inhibitory activity for the two proteins CDK- 1/PARP-1 and compare the results with the novel compound 645656
3- RMSD is not enough criteria to investigate the stability of the novel compounds, I suggest calculating the Root mean square deviation RMSF and the radius of gyration Rg for the novel compound and protein-ligand free and comparing the results, also calculating the average of the RMSD, RMSF, and Rg.
4- Authors add references under legends of the figures such as Figure 3,4,and, delete the reference from the legends.
5- In the X-axis in Figure 9, the author wrote Time (nsec), this is incorrect, the correct is ns, correct it.
Author Response
We would like to thank you Reviewer 2 for the constructive analysis of our manuscript.
We have followed, point-by-point, your suggestions, and we hope that the revised version could be reconsidered for submission to the International Journal of Molecular Sciences.
You can find below the responses:
The manuscript titled “In silico mixed ligand/structure-based design of new CDK- 1/PARP-1 dual inhibitors as anti-breast cancer agents” described two techniques, mixed ligand- and structure-based virtual screening which are used in identifying a novel compound known as 645656. Compound 645656 showed a significant affinity for both CDK-1 and PARP-1 enzymes which suggested its antiproliferative activity. Molecular dynamics simulations were carried out by the authors, to investigate the stability of the novel compounds within two proteins.
It is an interesting manuscript, the introduction is well-organized, the methods and results are well-represented, and the figures are clear and significant. My overall comment is accepting the manuscript after addressing some issues.
- Authors carried out molecular dynamics simulations for 20ns, which is not enough time for the system to reach equilibrium. I suggested carrying out MD for 150ns at least to investigate the stability of the compound.
Response: molecular dynamics simulations were performed for 100 ns for each analyzed complex. Since they demonstrated high stability already after the first 20-30 ns of the simulation time, for clarity we reported in the text the energy vS time graphs related to this part of the experiment only. In detail, complex 645656/CDK-1 reaches energy plateau after 20 ns, while for complex 645656/PARP-1 we extended the graphs until 30 ns to make more evident the energy plateau reached.
- Authors need to carry out molecular dynamics simulation for reference compounds that are known by their inhibitory activity for the two proteins CDK- 1/PARP-1 and compare the results with the novel compound 645656
Response: 100ns Molecular Dynamic Simulations for reference compounds olaparib and dinaciclib have been carried out on the corresponding target structures, PARP-1 and CDK-1, respectively. The results, including energy vS time graph, RMDS variation along time graph, the RMSF graph and the radius of gyration graph are shown in Supplementary Material. In detail, by a deep comparative analysis of all these output data related to both the reference compounds and the identified dual inhibitor 645656, it emerged a very close interaction pattern and comparable stability with both target proteins.
- RMSD is not enough criteria to investigate the stability of the novel compounds, I suggest calculating the Root mean square deviation RMSF and the radius of gyration Rg for the novel compound and protein-ligand free and comparing the results, also calculating the average of the RMSD, RMSF, and Rg.
Response: Protein Root Main Square Deviation, Ligand Root Main Square Deviation, Protein Root Main Square Fluctuation, Ligand Root Main Square Fluctuation, Radius of Gyration, and Binding Free Energy been computed for all the four complexes: 645656/CDK-1, 645656/PARP-1, olaparib/PARP-1, and dinaciclib/CDK-1. These results have been integrated in Supplementary Material.
- Authors add references under legends of the figures such as Figure 3,4, and delete the reference from the legends.
Response: References from figures 3-5,8 legends have been removed.
- In the X-axis in Figure 9, the author wrote Time (nsec), this is incorrect, the correct is ns, correct it.
Response: In the X-axis in Figure 9, “Time (nsec) has been corrected with “Time (ns)”.